# MERRF Mutation A8344G in a Four-Generation Family without Central Nervous System Involvement: Clinical and Molecular Characterization

**DOI:** 10.3390/jpm13010147

**Published:** 2023-01-11

**Authors:** Michela Ripolone, Simona Zanotti, Laura Napoli, Dario Ronchi, Patrizia Ciscato, Giacomo Pietro Comi, Maurizio Moggio, Monica Sciacco

**Affiliations:** 1Neuromuscular and Rare Diseases Unit, Department of Neuroscience, Foundation IRCCS Ca’ Granda Ospedale Maggiore Policlinico, 20122 Milan, Italy; 2Dino Ferrari Center, Department of Pathophysiology and Transplantation (DEPT), University of Milan, 20122 Milan, Italy

**Keywords:** MERRF, lipoma, central nervous system

## Abstract

A 53-year-old man approached our Neuromuscular Unit following an incidental finding of hyperckemia. Similar to his mother who had died at the age of 77 years, he was diabetic and had a few lipomas. The patient’s two sisters, aged 60 and 50 years, did not have any neurological symptoms. Proband’s skeletal muscle biopsy showed several COX-negative fibers, many of which were “ragged red”. Genetic analysis revealed the presence of the A8344G mtDNA mutation, which is most commonly associated with a maternally inherited multisystem mitochondrial disorder known as MERRF (myoclonus epilepsy with ragged-red fibers). The two sisters also carry the mutation. Family members on the maternal side were reported healthy. Although atypical phenotypes have been reported in association with the A8344G mutation, central nervous system (CSN) manifestations other than myoclonic epilepsy are always reported in the family tree. If present, our four-generation family manifestations are late-onset and do not affect CNS. This could be explained by the fact that the mutational load remains low and therefore prevents tissues/organs from reaching the pathologic threshold. The fact that this occurs throughout generations and that CNS, which has the highest energetic demand, is clinically spared, suggests that regulatory genes and/or pathways affect mitochondrial segregation and replication, and protect organs from progressive dysfunction.

## 1. Introduction

The A-to-G transition at nucleotide 8344 (m.8344A > G) of mtDNA is the prevalent mutation found in a multisystem disorder, and is known with the acronym MERRF (myoclonus epilepsy with ragged-red fibers). It is characterized by myoclonus, generalized epilepsy, ataxia, weakness, dementia as well as signs of multisystem involvement [1,2,3]. The histopathological study of the skeletal muscle tissue typically shows ragged-red fibers (RRFs) with the modified Gomori trichrome (MGT) stain and hyperactive fibers with the succinate dehydrogenase (SDH) stain. Histochemical reaction for cytochrome c oxidase (COX) shows lack of activity in RRFs and some non-RRFs [4,5,6]. Occasionally, RRFs may not be observed [7]. The presence of lipomas has often been reported in patients affected with MERRF and/or in their maternally-related family members [8,9,10].

Moreover, the m.8344A > G variant has been reported in association with isolated myopathy, lipomatosis with muscle lipid storage, or Leigh syndrome [11,12,13]. Other unusual manifestations include sudden infant death syndrome [14], spasmodic dysphonia [15], Parkinsonism with neuropathy and myopathy [16], infantile-onset ataxia, myoclonus and bilateral putaminal necrosis on brain MRI [17], sudden respiratory failure in adulthood [18], acute central and peripheral nervous system demyelinating disease [19], and typical MELAS (mitochondrial encephalomyopathy with lactic acidosis and stroke-like episodes) picture [20], reflecting the heterogenous clinical presentations associated resulting from mtDNA defects.

The pathologic mutation affects all tissues; however, since mtDNA mutations are heteroplasmic, the variable tissue distribution of mutated mtDNA usually occurs in the same individual. The m.8344A > G variant is usually heteroplasmic in tissues collected from classical MERRF patients and the level of mutational load required to display a biochemical phenotype (biochemical threshold) is in the range between 60 and 90%, suggesting a moderate detrimental behavior for this nucleotide change. Both the heteroplasmy and the selective tissue vulnerability to impaired oxidative metabolism (skeletal/cardiac muscle and brain have a higher energetic demand) are important factors in determining the clinical expression of mtDNA mutations. These aspects, along with the different regional levels of mutant DNA, the compensatory increase in global mtDNA content, and the presence of nuclear modifiers, hamper the establishment of a clear correlation between the genotype and clinical phenotype [21,22].

Although there may be a high clinical variability among members of the same family, central nervous system (CNS) manifestations have always been reported throughout family generations. In this study, we describe a patient carrying the m.8344A > G mutation in mitochondrial DNA, presenting with late onset myopathy, multiple lipomas, and diabetes.

Neither the proband nor the other affected family members show signs of CNS involvement.

## 2. Case Report

A 53-year-old man approached our Neuromuscular Unit due to an incidental finding of hyperckemia (between 300 and 400 U/L, n.v. < 180 U/L). He was diabetic and had a few lipomas, but otherwise asymptomatic. Almost 2 years later, he developed fatigue which progressively worsened. CK levels had moderately increased to 970 U/L. The neurological examination was normal and his past medical history was unremarkable.

The patient’s mother, who had died at 77 years of age, was diabetic, cardiopathic, and displayed multiple lipomas. Proband’s sisters, aged 60 and 50 years, presented normal serum CK levels. The elder sister has mild bilateral eyelid ptosis and had suffered from thyroiditis; while the younger sister had undergone surgery for colon cancer at the age of 48 years and the follow-up was negative. 

The first sister has three children, two males and one female, aged 33, 23, and 32 years, respectively. The 33-year-old son has two children, one male and one female. The 32-year-old daughter has a son. The second sister has a 13-year-old daughter and a son aged 11 years. They are all reported asymptomatic (Figure 1A). Descendants of two maternal aunts are reported healthy and did not undergo any genetic evaluation.

After an EMG examination, which showed a myopathic pattern in proximal four-limb muscles, the proband underwent left biceps skeletal muscle biopsy. Following plain brain CT scan (Figure 1D) and EEG, both examinations were normal.

Moreover, we evaluated both sisters whose neurological examination was normal except for the mild eyelid ptosis in the elder one. Blood and urinary samples were taken from both subjects for DNA extraction.

## 3. Materials and Methods

After the patient had signed a written informed consent on 28 February 2005, a skeletal muscle specimen (biopsy code number: 96974) from his left biceps brachii muscle was obtained by an open biopsy, according to a protocol approved by the Institutional Review Board of the “IRCCS Ca’ Granda Foundation Ospedale Maggiore Policlinico, Italy”.

A total of 8 µm-thick cryostatic sections were processed according to standard histological and histochemical techniques [23]. Mitochondrial enzymatic activity was demonstrated by COX, SDH, and double COX-SDH staining [23]. Enzymatic evaluations for acid phosphatase (AP), phosphofructokinase (PFK), myophosphorylase (PYGM), and myoadenylate deaminase (MAD) were also performed.

Immunohistochemistry for sarcolemmal proteins (dystrophin, alpha-, and gamma-sarcoglycan) and for an evaluation of possible inflammatory signs were performed [24].

Furthermore, after obtaining written consent, genomic DNA was extracted from peripheral blood, urine, and muscle from both the proband (muscle, blood, urine) and his sisters (blood, urine). The extracted mtDNA was PCR-amplified using MitoSEQ Resequencing System (Applied Biosystem, FosterCity, CA, USA) and sequenced on an ABI PRISM 3100 Genetic Analyzer (Applied Biosystem).

Mutational loads in the patient’s tissues were assessed by PCR-RFLP performed using a modified primer that creates a *BglI*-restriction site in mutant molecules. Aliquots of PCR products were digested and electrophoresed on a 4% agarose gel. The proportion of mutant mtDNA was evaluated by densitometry using the NIH ImageJ 2.1.0 software (NIH National Institute of Health, Bethesda, MD, USA) (https://imagej.nih.gov/ij/download.html, accessed on 5 November 2022) in the month of September 2011.

## 4. Results

Sections stained with MGT showed quite a few ragged-red fibers. Histochemical reactions for mitochondrial enzymatic activity showed several COX-negative fibers, many of which were intensely stained with SDH (RRFs) (Figure 1C). Glycogen content was normal and no lipid storage was present.

Enzymatic activities for PKF, PYGM, and MAD were normal. The staining for PA showed a slightly increased subsarcolemmal signal in a few fibers. Immunohistochemistry for sarcolemmal proteins was normal (data not shown).

Genetic analysis of the entire mtDNA sequence in proband’s muscle revealed the presence of the A8344G mtDNA mutation. The same mutation was detected in the two sisters.

The degree of heteroplasmy of this mutation was analyzed in proband’s skeletal muscle and in blood leukocytes and urinary sediment samples from both the proband and his sisters. Densitometric analysis in the patient’s tissues revealed that the A8344G mutation accounted for 25.2% of the total mtDNA in muscle, 60% in blood, and 61% in urine (Figure 1B). The younger sister presented a mutational load of 35% in blood and 18% in urine, while the elder sister had 12.1% mtDNA mutation in blood, but no mutation was detected in urine. Detailed quantification of the mutational load in the family members is presented in Figure 1B.

## 5. Discussion

MERRF syndrome is a devastating neuromuscular disorder characterized by myoclonic epilepsy, generalized weakness, muscle wasting, cerebellar ataxia, deafness, dementia, and RRFs at muscle biopsy. It is transmitted through maternal lineage [1,2,5]. About 80% of MERRF cases are caused by the A8344G mutation in the tRNA Lys gene, [3,25] although a few less frequent mtDNA point mutations have also been found in MERRF patients [26,27,28,29,30].

Our patient did present RRFs and lipomas, the latter was also diagnosed in his mother. These findings, along with the presence of other features indicating multisystem involvement in both patient and family (diabetes, cardiopathy, endocrine dysfunction) as well as lack of lipid storage at muscle biopsy, prompted a diagnosis of atypical MERRF syndrome.

Our patient showed an atypical clinical presentation, with isolated hyperckemia at onset followed by development of myopathy in his fifties. The associated presence of maternally inherited diabetes and lipomas suggested a diagnosis of mitochondrial disorder, which was confirmed by both morphological and biomolecular findings.

Lipomas have often been reported in patients bearing the A8344G mutation in association with MERRF syndrome or other central nervous system involvement [8,9,10,31,32]. Indeed, the presence of maternally inherited lipomas associated with the involvement of other organs/systems is an almost unequivocal indication of the presence of mutations in the tRNA Lys. It is not known how an impaired mitochondrial function, due to mutations in tRNA Lys, causes this effect on adipose tissue; however, there is evidence that mitochondrial function is important for normal development of adipose tissue in humans [33].

The A8344G mutation has been considered a relatively “benign” mutation, since a high degree of mutational load is required to produce clinical manifestations. In skeletal muscle tissue, the threshold level beyond which the pathological phenotype becomes evident is estimated to be higher than 60% mutational load [34]. Indeed, it has been suggested that approximately 15% of residual wildtype mtDNA is sufficient to restore translation and COX activity to near-normal levels, thus “rescuing” the clinical phenotype. Interestingly, our patient had a low mutational load in muscle (25%), and this can explain the late onset of symptoms.

Furthermore, the A8344G mutation is usually present in high proportion in DNA from urine and blood, the mutational load being ordinarily higher in urine than in blood [35]. In accordance with the data reported in the literature, our patient has a mutational load of 61% in urine and 60% in blood; however, both his sisters have a higher mutant load in blood than in urine. We were unable to establish the mutational load in the lipomas since the patient refused to undergo lipoma biopsy.

In our family, we established a certain positive correlation between the severity of clinical signs and instrumental evidence. Indeed, only the proband, who, unlike his sisters, has increased serum CK levels, is symptomatic. We could not make any correlations in terms of skeletal muscle mutational load since the two sisters did not undergo skeletal muscle biopsy [31,32,33].

Muscle biopsy showed typical histopathological features of MERRF [5,6,36], in particular the presence of RRFs at MGT, confirmed by increased SDH activity, the absence of COX activity in most RRFs, and a number of COX-negative/deficient non-RRFs.

The absence of central nervous system involvement is a peculiar feature in our family since no other MERRF families without any central nervous system involvement have been reported to date. A possible explanation is that the mutational load remains low, especially in CNS, and prevents tissues/organs from reaching the pathologic threshold. The fact that this occurs throughout generations and that the tissue with the highest energetic demand is clinically spared, suggests that regulatory genes and/or pathways affect mitochondrial segregation and replication, and protect organs from progressive dysfunction. As suggested by Letrit et al., a lack of correlation between the degree of mtDNA heteroplasmy and clinical symptoms related to a particular organ can indicate the presence of tissue-specific nuclear factors that modify the phenotypic expression of the A8344G mutation, or, perhaps rather than a specific nuclear factor, there are merely tissue differences in the requirements for the particular subunit of the respiratory chain involved [21].

## 6. Conclusions

In conclusion, our report highlights the broad clinical spectrum of MERRF syndrome, which can also occur as a pure myopathy. Given the large number of atypical cases, we would like to emphasize that it is easy to underestimate progressive and even potentially invalidating diseases.

The combination of a high clinical suspect, histological, molecular genetics, and biochemical investigation remains essential for the diagnosis of MERRF.

## Figures and Tables

**Figure 1 jpm-13-00147-f001:**
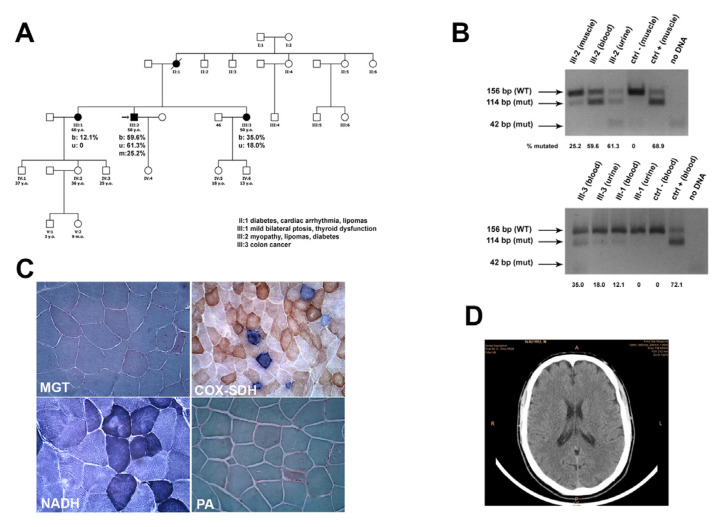
(**A**) Pedigree of family. (**B**) PCR-RLFP analysis of amplicons obtained from muscle, urine, and blood of the indicated subjects, electrophoresed on 4% agarose gel after *BglI* cut. The numbers under each lane indicate the estimated mutational load in the sample. (**C**) Morphological examination showed some RRFs at MGT, several COX-negative fibers, many of which are intensely SDH-positive (RRFs). Scale bar 50 μm. (**D**) Brain CT scan.

## Data Availability

The data for this article are not publicly available to ensure patient anonymity. Requests to access the data should be directed to the corresponding author.

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
