# Peer review of "MERRF Mutation A8344G in a Four-Generation Family without Central Nervous System Involvement: Clinical and Molecular Characterization"

_jpm, 2023, doi:10.3390/jpm13010147_

Round 1
Reviewer 1 Report
Dear authors
thank you for reporting this very interesting family with a8344G mutation.
In the title , the introduction and conclusion you focused on the MERRF syndrome while it was more interesting to describe the different cases reported with the a8344G mutation like leigh, myopathy, stroke like...and to strat from line 54 and then described the MERRF syndrome with a8344G mutation....especially that none patient in this family had central nervous symptoms or epilepsy (Neither the proband nor the other affected family members show signs of CNS 77 involvement.)line 77.
It is difficult to understand how this patient was diagnostic with MERRF csyndrome line 161:"Atypical clinical phenotypes have been attributed to the A8344G mtDNA mutation 161 [27, but, to our knowledge, there are no reported MERRF cases without any central 162 nervous system involvement among family members."
On the methodology you give informations about cousins but there is no information about clinical or genetic results of them.
Several references are related to mitochondrial disease, and some references on atypical presentation of a8344G mutation are lacking, such as: A8344G mitochondrial DNA mutation with typical mitochondrial encephalomyopathy with lactic acidosis and stroke-like episodes syndrome : Ildikó Vastagh, Anikó Gál, Viktória Reményi, Judit Semjén, Tímea Lukács, Attila Valikovics, Mária Judit Molnár
Author Response
Dear authors
thank you for reporting this very interesting family with a8344G mutation.
In the title, the introduction and conclusion you focused on the MERRF syndrome while it was more interesting to describe the different cases reported with the a8344G mutation like leigh, myopathy, stroke like...and to start from line 54 and then described the MERRF syndrome with a8344G mutation.... especially that none patient in this family had central nervous symptoms or epilepsy (Neither the proband nor the other affected family members show signs of CNS 77 involvement) line 77. It is difficult to understand how this patient was diagnostic with MERRF syndrome line 161:"Atypical clinical phenotypes have been attributed to the A8344G mtDNA mutation 161 [27], but, to our knowledge, there are no reported MERRF cases without any central 162 nervous system involvement among family members."
We do thank the reviewer for this observation which is indeed very pertinent given the absence of myoclonus in our patient. We did diagnose an atypical MERRF syndrome because of the association of RRFs and maternally inherited lipomas in addition to signs of multisystem involvement in both patient and family members (diabetes, cardiopathy, endocrine dysfunction). In mitochondrial dysfunction, lipomas are mostly described in typical MERRF patients with the exception of few A8344G cases in which patients had a lipomatosis associated with isolated proximal myopathy characterized by lipid storage at muscle biopsy and respiratory chain complexes deficiency, both absent in our patient (Munoz-Malaga et al. Muscle Nerve 2000).
To better explain this relevant point, we modified the text, more precisely, the first part of the Introduction (lines 54-63, 65, 69-71), the muscle biopsy Results (line 142-143) indicating that there was no lipid storage, and the Discussion (lines 174-178, 186-188, 215). We also modified the line 23 in Abstract.
According with the changes made in the text, we have added 4 new references (8, 9, 10) and eliminated 5 other references previously indicated: 8, 15, 16, 18, 20.
On the methodology you give information about cousins but there is no information about clinical or genetic results of them.
All cousins on the maternal side (descendants of two maternal aunts) were clinically unaffected and did not undergo any genetic evaluation. The latter information has been added to the text (line 108).
Several references are related to mitochondrial disease, and some references on atypical presentation of a8344G mutation are lacking, such as: A8344G mitochondrial DNA mutation with typical mitochondrial encephalomyopathy with lactic acidosis and stroke-like episodes syndrome : Ildikó Vastagh, Anikó Gál, Viktória Reményi, Judit Semjén, Tímea Lukács, Attila Valikovics, Mária Judit Molnár
We apologize for not including this peculiar reference reporting a clinically MELAS picture with a A8344G mutation. The reference has now been added and quoted in line 71 where atypical A8344G phenotypes are mentioned (Ref 20).
Reviewer 2 Report
I really have only a few very minor matters to raise.
· I wondered whether there was any CT evidence of cerebellar atrophy since long-standing and not progressive cerebellar atrophy may be clinically silent.
· Were there any electrocardiogram changes?
· Were nerve conduction velocities measured in conjunction with the EMG and, if so, were the velocities normal?
Author Response
I really have only a few very minor matters to raise.
- I wondered whether there was any CT evidence of cerebellar atrophy since long-standing and not progressive cerebellar atrophy may be clinically silent.
There was no brain CT evidence of cerebellar atrophy
- Were there any electrocardiogram changes?
The electrocardiogram showed a regular sinus rhythm
- Were nerve conduction velocities measured in conjunction with the EMG and, if so, were the velocities normal?
Yes, both motor ad sensory nerve conduction velocities were measured and they were normal